# Antimicrobial prophylaxis does not improve post-surgical outcomes in SIV/SHIV-uninfected or SIV/SHIV-infected macaques (*Macaca mulatta* and *Macaca fascicularis*) based on a retrospective analysis

**Cassandra Moats**[1,2][☯]*, **Kimberly Cook**[1][☯], **Kimberly Armantrout**[1‡], **Hugh Crank**[1‡], **Samantha Uttke**[1‡], **Kelly Maher**[1‡], **Rachele M. Bochart**[1‡], **George Lawrence**[3][☯], **Michael K. Axthelm**[1‡], **Jeremy V. Smedley**[1‡]

1 Infectious Disease Resource, Oregon National Primate Research Center, Beaverton, Oregon, United States of America, 2 Department of Molecular and Comparative Pathobiology, Johns Hopkins School of Medicine, Baltimore, Maryland, United States of America, 3 Director's Office, Oregon National Primate Research Center, Beaverton, Oregon, United States of America

☯ These authors contributed equally to this work.
‡ KA, HC, SU, KM, RMB, MKA, and JVS also contributed equally to this work.
* cmoats1@jhmi.edu

**Data Availability Statement:** All relevant data are within the paper.

## Abstract

Surgical antimicrobial prophylaxis is indicated when performing contaminated surgeries, when specific surgical implants are placed, and for prolonged surgical procedures. Unnecessary prophylactic antibiotics are often utilized for macaque surgeries, despite medical and veterinary guidelines. In this study we compared complication rates in macaques receiving peripheral lymph node (PLN) and laparoscopic biopsies, with and without antimicrobial prophylaxis. A majority of animals were SIV or SHIV infected at the time of surgery, so we also compared post-operative complication rates based on infection status. We found no significant difference in PLN biopsy complication rates for animals that received antimicrobial prophylaxis versus those that did not. Animals who underwent laparoscopic procedures and received prophylactic antibiotics had a higher complication rate than those who did not receive them. Complication rates did not differ significantly for SIV/SHIV infected versus uninfected animals for both laparoscopic biopsy procedures and PLN biopsy procedures. SIV/SHIV infected animals that underwent PLN biopsies had no significant difference in complication rates with and without antimicrobial prophylaxis, and SIV/SHIV infected animals receiving prophylactic antibiotics for laparoscopic biopsies had a higher complication rate than those that did not. This study suggests that perioperative prophylactic antibiotics have no role in the management of SIV/SHIV-infected and uninfected macaques undergoing clean, minimally invasive surgeries. Additionally, we recommend eliminating unnecessary antibiotic use in study animals due to their potential confounding impacts on research models and their potential to promote antimicrobial resistance.

**Funding:** This work was supported by the Office of the Director, National Institutes of Health (3U42 OD023038), and the Oregon National Primate Research Center NIH Core Grant (P51OD011092) The funders had no role in study design, data collection and analysis, decision to publish, or preparation of the manuscript.

**Competing interests:** The authors have declared that no competing interests exist.

## Introduction

Surgical antimicrobial prophylaxis is the use of a very brief antimicrobial agent initiated immediately prior to an operation [1]. The 2017 CDC Guideline for the Prevention of Surgical Site Infection recommends that prophylactic antibiotics be administered only when indicated, based on published clinical practice guidelines, and timed such that the bactericidal concentration of the agent(s) is established in the serum and tissues when the incision is made [2]. They also emphasize that in clean and clean-contaminated procedures, additional prophylactic antimicrobial doses should not be provided after the incision is closed [2], as there are no identified randomized controlled trials that show benefits of parenteral antimicrobial prophylaxis and its effect on the risk of surgical site infections (SSI). Both medical and veterinary guidelines on surgical antimicrobial prophylaxis stress that clean and clean-contaminated surgical wounds do not require ongoing antimicrobial therapy unless there is a break in sterile technique, a specific surgical implant has been placed, or the surgery lasts an extended period of time [1, 3, 4]. Host factors, such as immune status, may also determine if antimicrobial prophylaxis is extended into the post-operative period [5].

Non-human primates (NHP) models are commonly utilized in studies of infectious disease, neuroscience, behavior, and reproduction. Surgeries are performed in NHPs for both clinical and research purposes, and many NHPs have multiple surgeries during their research protocol. There is a perception among researchers and veterinary staff that NHPs require prophylactic antibiotics for surgical procedures, regardless of wound classification, surgery length, host status, or implant placement. This perception arises from the belief that NHPs tend to pick at surgical incisions, and given their environment and grooming tendencies, have the potential to carry fecal material into the incision. Thus, at some institutions, antimicrobial prophylaxis is routinely provided to animals to prevent SSIs that could occur due to animal manipulation of incisions (unpublished information; author's experience). This practice is in opposition to current recommendations published by authorities in both human and veterinary medicine [1, 2].

The Simian Immunodeficiency Virus (SIV)-infected macaque is the leading animal model of HIV, as the model closely recapitulates the pathogenesis of HIV in humans [6]. SIV-infected animals commonly undergo serial surgical biopsies throughout the course of their study to sample primary and secondary lymphoid and mucosal sites. Common surgical procedures include peripheral lymph node biopsy, laparoscopic biopsies (mesenteric lymph node, liver, and spleen), and mucosal pinch biopsies [7–9]. Unfortunately, antimicrobial prophylaxis for these procedures is frequently unreported in the literature, despite their common use for the reasons previously discussed and due to the perceived increased risk of SSIs from presumed SIV related immunosuppression [10]. In one publication describing a vaccination strategy to prevent SIV in rhesus macaques, animals received prophylactic antibiotics for an undefined period following lymph node and rectal biopsies [11]. In another study, the authors provided peri-operative antibiotics to HIV-1-infected pigtailed macaques following serial laparoscopic procedures [10]. However, many other similar publications do not cite prophylactic antimicrobial use, which is not surprising, as other perioperative details such as anesthesia and analgesia are also not described [7, 12]. Therefore, based on the authors' experiences, it is possible that during many of these studies, prophylactic antibiotics were administered in the peri-operative period but were not described in the materials and methods sections. Similarly, use of antimicrobial prophylaxis is inconsistent in the literature for peripheral lymph node biopsies in HIV infected patients [13, 14].

Immunodeficiency from HIV can increase the risk of SSIs and other post-operative complications, particularly in the presence of AIDS [5, 15, 16]. However, peri-operative prophylactic

antibiotics are not considered necessary for clean procedures in HIV-positive patients, such as lymph node biopsy [13, 14]. In a study examining rates of postoperative complications among HIV positive women undergoing obstetric and gynecologic procedures, there were no differences in complication rates between HIV-positive patients receiving peri-operative antimicrobial prophylaxis and those who did not [15]. In fact, some studies show that antibiotic prophylaxis increases the risk of SSIs [3, 16]. Thus, we can assume from experiences in HIV-infected individuals that SIV-infected macaques undergoing similar clean procedures likely do not require prophylactic antibiotics, and that prophylactic antimicrobial use could possibly increase the risk of SSI in the post-operative period.

The purpose of this study was to compare post-operative complications following common research surgeries in macaques receiving peri-operative prophylactic antibiotics with macaques that did not receive them. We also examined the effect of SIV and Simian-Human Immunodeficiency Virus (SHIV) status on complication rates, with and without the use of prophylactic antibiotics. We hypothesized that there would be no difference in complication rates between animals that received prophylactic antibiotics versus those that did not. We also hypothesized that there would be no difference between complication rates in SIV or SHIV infected animals, with and without peri-operative prophylactic antibiotics.

## Materials and methods

### Animal information and ethics statement

All animals were used in accordance with the policies of the Institutional Animal Care and Use Committee (IACUC) at the ONPRC, an AAALAC-accredited institution, which abides by the USDA Animal Welfare Regulations and the Guide for the Care and Use of Laboratory Animals. All enrolled macaques were SPF (serologically negative for simian T-lymphotropic virus-1, SIV, simian type D retrovirus, and Macacine herpes 1) prior to the start of the study, and were captive-born at ONPRC. All animals inhabited socially-housed indoor/outdoor enclosures before their transfer to indoor standardized housing prior to study start, and were singly or socially-housed during the study in accordance with the IACUC protocol. Throughout the study, all animals were uniformly fed Purina LabDiet 5000 (Purina Mills International, St. Louis, MO), daily nutritional enrichment items (grains, fruits, or vegetables) and had ad libitum access to water. In addition to social housing and nutritional enrichment, animals also received environmental enrichment in the form of toys, music/radio, and television daily. All animals were observed for health or behavioral concerns at minimum twice daily by research, veterinary, and/or husbandry staff. Animals requiring euthanasia for study or clinical endpoint purposes were sedated with an intramuscular injection of 20 mg/kg ketamine HCl (Ketathesia™, Henry Schein Animal Health) followed by an overdose of intravenous pentobarbital solution, in accordance with the AVMA Guidelines for the Euthanasia of Animals. However, as this study was a retrospective records review of animals undergoing surgical procedures for various studies conducted by the ONPRC IDR, no animals required euthanasia directly as a result of this study. Anesthesia and analgesia details are described in detail in the subsequent subsections.

### Peripheral lymph node biopsy procedure

Animals were fasted overnight and sedated with an intramuscular injection of 10–12 mg/kg ketamine HCl (Ketathesia™, Henry Schein Animal Health) and 0.015 mg/kg dexmedetomidine hydrochloride (Dexmedesed™, Dechra, Overland Park, KS). Some animals were also intubated and placed on isoflurane (0.8–2%) if they received mucosal or laparoscopic biopsy procedures in addition to PLN biopsy. Once anesthetized, all animals received an intramuscular (IM)

injection of buprenorphine HCl (Buprenex®, Reckitt Benckiser, Slough, Berkshire, England; 0.3 mg/animal weighing > 3 kg; 0.15 mg/animal 1.5–3 kg; 0.04 mg/kg/animal <1.5 kg) or a subcutaneous (SC) injection of sustained release buprenorphine (Buprenorphine SR, Zoo-Pharm, Fort Collins, CO; 0.2 mg/kg/animal). Ophthalmic ointment was applied to the corneas. PLN sites (axillary, inguinal, or submandibular) were shaved, and animals were laid in dorsal recumbency on a heating implement (water recirculating blanket, heating pad, or heated table). Heart rate and peripheral oxygen saturation were continuously monitored throughout the procedure. Limbs or head/neck were positioned with ties or tape to facilitate surgical access, and surgical sites were aseptically prepared for surgery. Aseptic preparation involved either three alternating scrubs of betadine (Povidone Iodine Swabsticks, Aplicare®, Meriden, CT) and 70% alcohol, three alternating scrubs of 2% chlorhexidine solution (VetOne®, Boise, ID) and 70% alcohol, or Chloraprep (Chloraprep® 3 mL applicator, CareFusion, San Diego, CA) application per manufacturer's instruction.

Once prepared for surgery, sites were draped with a transparent ophthalmic adhesive drape (Steri-Drape™, 3M, Maplewood, MN), and all procedures were performed using aseptic technique. Incisions were made with a #15 surgical blade directly over the proposed lymph node biopsy site, and a combination of blunt and sharp dissection was used to isolate a group of lymph nodes. Prior to excision, lymph node pedicles were ligated with 4–0 absorbable monofilament suture. Following excision, the biopsy site was examined for hemostasis before releasing the pedicle. Biopsy sites were closed in 2–3 layers depending on location, using 4–0 absorbable monofilament suture. Axillary and inguinal sites were closed in two layers (subcutaneous and skin), while submandibular sites were closed in three layers (muscle, subcutaneous and skin). All skin closure was performed using an intradermal pattern, followed by application of cyanoacrylate tissue adhesive (Vetbond™ Tissue Adhesive, 3M, Maplewood, MN). Instillation of local anesthesia along the incision, usually with bupivacaine, was also performed. Following the procedure, all animals were reversed with atipamezole hydrochloride (Antisedan®, Zoetis, Kalamazoo, MI), extubated if indicated, and recovered. In the uncommon case where an animal had a local reaction to SC sustained release buprenorphine, IM buprenorphine HCl was utilized, and animals received 48 hours of analgesia (dosed every 12 hours). Animals that received sustained release buprenorphine did not receive additional doses unless additional analgesia was required.

## Laparoscopic biopsy procedure

Animals were fasted overnight and sedated with an IM injection of 10 mg/kg ketamine HCl (Ketathesia™, Henry Schein Animal Health) with or without 0.015 mg/kg dexmedetomidine hydrochloride (Dexmedesed™, Dechra, Overland Park, KS). Following sedation, animals were intubated and placed on isoflurane (0.8–2%). All animals received analgesic, as described above, and ophthalmic ointment was applied to the corneas. The abdomen was shaved, and animals were laid in right dorsolateral recumbency on a heating implement (water recirculating blanket, heating pad, or heated table). Heart rate, ECG, respiration rate, end-tidal $CO_2$, peripheral oxygen saturation, and temperature were continuously monitored throughout the procedure, and animals received intravenous isotonic fluids at a rate of 10 ml/kg/hr. Limbs were secured with ties to facilitate surgical access, and the abdomen was aseptically prepared for surgery using a Chloraprep applicator (Chloraprep® 10.5 mL applicator, CareFusion, San Diego, CA) per manufacturer's instruction.

Once prepared for surgery, the patient was draped with sterile towels and cloth drapes, and all procedures were performed using aseptic technique. Laparoscopic MLN and liver biopsies were otherwise collected as described by Zevin *et al* [9]. If the spleen was biopsied, the cannula

and endoscope were relocated to the paraumbilical port site, and the spleen was visualized. Clamshell biopsy forceps (5 mm) were used to biopsy the margin of the spleen. Following the procedure, all animals were reversed with atipamezole hydrochloride (Antisedan®, Zoetis, Kalamazoo, MI), extubated, and recovered. In the uncommon case where an animal had a local reaction to SC sustained release buprenorphine, IM buprenorphine HCl was utilized, and animals received 48 hours of analgesia (dosed every 12 hours). Animals that received sustained release buprenorphine did not receive additional doses unless additional analgesia was required.

## Administration of perioperative prophylactic antibiotics

Animals that received perioperative prophylactic antibiotics received cefazolin 25 mg/kg IM once, before the first incision, usually during animal preparation for surgery. Prophylaxis was continued into the post-operative period for 3–5 days, based on the ONPRC requirements. During the post-operative period, animals received either cefazolin 25 mg/kg IM twice daily, or cephalexin 25 mg/kg per os (PO).

## Determination of SIV/SHIV status

Animals undergoing surgical procedures were assigned to the SIV/SHIV uninfected status or SIV/SHIV infected status based on the infection status designated in their medical record, with any prior or concurrent positive PCR result indicating that they were infected at the time of the procedure. Infection status was determined by a positive quantitative real-time PCR outcome as previously described [17].

## Data collection and identification of complications

Surgical records from macaques undergoing PLN and/or laparoscopic biopsy from May 1st, 2018 through January 19th, 2021 were extracted from our institutional electronic health records (EHR) system. For this study, demographic and clinical data including age and body condition score (BCS) at time of surgery, species, SIV/SHIV infection status, surgical procedure, use of perioperative medications, use of prophylactic antibiotics, and post-operative observation duration were queried directly or computed as required.

In order to address our question regarding surgical complications, we required means to flag complications based on criteria including: prescription of antibiotics (cefazolin or cephalexin) from days 1 through 8 post-operatively; prescription of non-steroidal anti-inflammatory drugs (NSAIDs) from days 1 through 8 post-operatively; prescription of additional buprenorphine HCl or buprenorphine sustained release doses from days 1 through 8 post-operatively; surgical cases which required post-operative observation for 9 or more days (standard duration 7–8 days). Many of these variables cannot be queried directly from or computed with the database application in a practical way due to 1) complexity of the required code required and 2) the resulting computation time. In order to overcome these limitations of conventional database applications, we've developed a computational platform using functional programming technology that allows us to work with research datasets. This computational integration platform is written in the Wolfram Language (Wolfram Mathematica® 9.0.1, Wolfram Research Inc, Champaign IL).

Procedures flagged for potential post-operative complications underwent further clinical records review to confirm the post-operative complication status. Records review included: review of post-operative NSAID, opiate, and antibiotic use to confirm prescription for surgical complications, rather than clinical concerns unrelated to the procedure; review of cases with prolonged post-operative observation periods to determine reason for prolonged observation;

and review of all post-operative case observation notes. Case observation notes which represented possible post-operative complications included: inappetence; moderate to severe inflammation, swelling or bruising; evidence of pain; incision discharge; surgical site abscessation; suture removal; incision ulceration; and/or incision dehiscence. From the records review, we considered a true post-operative complication as one that either required medical or surgical treatment, and/or resulted in the need to increase the standard post-operative observation period to allow for further monitoring to ensure the animal did not require further treatment. Following identification of true complications, complication rates were assessed for all procedures. Complication rates for each condition (prophylactic antibiotics versus no prophylactic antibiotics) were calculated by dividing the number of complications for each procedure (either peripheral lymph node biopsy or laparoscopic biopsy) by the number of total procedures performed.

## Statistical analysis

We used a series of randomization tests (also known as a permutation test) to determine if the rate of surgical complications differed between antibiotic-treated and untreated macaques and to determine if the rate of complications differed by SIV/SHIV infection status. We used the difference in the proportion of animals that had surgical complications between treatment groups in our observed data as our test statistic. This test statistic was compared to an empirical null distribution built by resampling our data to determine statistical significance. Complications were permuted 50,000 times to shuffle the association with prophylactic treatment [18]. We calculated the difference in the proportion of complications between groups for each of these permutations to represent a null distribution with no association between treatment status and surgical complication outcome. The proportion of permuted values that were as large or larger than the observed difference in complication rates between groups was used to represent a non-parametric p-value. Analyses were performed using R statistical Software (R Core Team, 2020) [19]. Results were considered statistically significant when the $p$ value was less than 0.05.

## Results

### Demographics

From May 1st, 2018 through January 19th, 2021, 3,629 surgeries were performed for various infectious disease studies in 1,056 macaques, which included 2,230 PLN (axillary, inguinal, and/or submandibular) and 1,399 laparoscopic biopsies (MLN, liver, and/or spleen). Most animals received multiple PLN and laparoscopic biopsies during the time period examined. Animal demographic information is provided in Table 1. Retrospective study group assignment is summarized in Fig 1.

### Complication rates: Prophylactic vs no prophylactic antibiotics

Randomization tests were performed for total complication rate comparing PLN and laparoscopic biopsies where animals received prophylactic antibiotics (PLN $n$ = 1,011 procedures, laparoscopic $n$ = 83 procedures), and biopsies where they did not (PLN $n$ = 1,219 procedures, laparoscopic $n$ = 1,316 procedures). There was no significant difference in PLN biopsy complication rates with (2.37%) and without (2.05%) prophylactic antibiotics ($p$ = 0.341). Laparoscopic biopsy complication rates were significantly higher for animals that received prophylactic antibiotics (3.61%) than those that did not (0.76%; $p$ = 0.038). Total complication rates are shown in Table 2.

**Table 1. Study animal demographics information.**

| Characteristic | |
|---|---|
| Species | Rhesus macaque $n$ = 1,006; Cynomolgus macaque $n$ = 50 |
| Age range | 0.04–25.56 years old |
| Body condition score (BCS) range | 1.5–5.0 |
| Sex | Male $n$ = 616; Female $n$ = 440 |
| Infection status at time of peripheral lymph node biopsy | SIV/SHIV infected $n$ = 1,340 procedures; uninfected $n$ = 890 procedures |
| Infection status at time of laparoscopic biopsy | SIV/SHIV infected $n$ = 861 procedures; uninfected $n$ = 538 procedures |

Demographics information for macaques assigned to this retrospective study. Note that most animals received multiple PLN and laparoscopic biopsies during the time period examined.

## Complication rates: Infection status

Randomization tests were performed for total complication rates comparing PLN biopsy procedures in SIV/SHIV infected and uninfected animals, with and without use of prophylactic antibiotics (infected, PLN with prophylaxis $n$ = 455 procedures; infected, PLN without prophylaxis $n$ = 885; uninfected, PLN with prophylaxis $n$ = 556; uninfected, PLN without prophylaxis $n$ = 334). There was no significant difference in PLN biopsy complication rates in SIV/SHIV infected (2.31%) and uninfected (2.02%) animals ($p$ = 0.375). Additionally, there were no significant differences in complication rates for PLN biopsy procedures in SIV/SHIV infected animals with (2.64%) and without (2.15%) the use of antimicrobial prophylaxis ($p$ = 0.249). Similarly, SIV/SHIV uninfected animals that underwent PLN biopsies and received prophylactic antibiotics had no significant differences in complication rates (2.16% vs 1.80%; $p$ = 0.250). Complication rates for PLN biopsies based on SIV/SHIV status are shown in Table 3.

Randomization tests were also performed for total complication rates comparing the same conditions for laparoscopic biopsy procedures (infected, laparoscopy with prophylaxis $n$ = 39 procedures; infected, laparoscopy without prophylaxis $n$ = 822; uninfected, laparoscopy with prophylaxis $n$ = 44; uninfected, laparoscopy without prophylaxis $n$ = 494). SIV/SHIV infected animals undergoing laparoscopy had a lower complication rate (0.46%) than uninfected animals (1.67%; $p$ = 0.995), but this did not differ significantly. SIV/SHIV infected animals undergoing laparoscopic biopsies had a significantly higher complication rate when receiving prophylactic antibiotics (5.13%) versus when they did not (0.24%; $p$ = 0.011). Similarly, uninfected animals had a higher complication rate when receiving antimicrobial prophylaxis

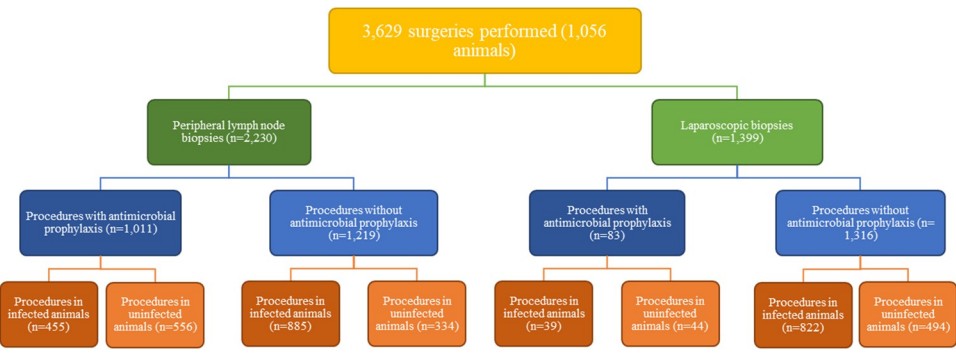

**Fig 1. Flow chart for retrospective study group assignment.**

**Table 2. Surgical complication rates: Prophylactic vs no prophylactic antibiotics.**

| Surgery Type | Prophylaxis | No Prophylaxis | Total |
|---|---|---|---|
| Peripheral lymph node biopsies | 2.37% (n = 1,011) | 2.05% (n = 1,219) | 2.20% (n = 2,230) |
| Laparoscopic biopsies | 3.61%* (n = 83) | 0.76%* (n = 1,316) | 0.93% (n = 1,399) |

Complication rates of peripheral lymph node (PLN) and laparoscopic biopsy (mesenteric lymph node, liver, and/or spleen) procedures in all animals, with and without the use of prophylactic antibiotics. Asterisk (*) indicates statistically significant results ($p$ value <0.05).

(2.27%) compared to when they did not (1.62%; $p = 0.382$), though this difference was not significant. Complication rates for laparoscopic biopsies based on SIV/SHIV status are shown in Table 3.

## Discussion

Antibiotics can induce colitis and affect key immune parameters, such as increasing the numbers of significant target cells (Th17+ CD4 T cells) for SIV infection in the colon, which could negatively influence infectious disease models [8, 20]. Based on this data, we convinced our institution to change its practice and allow common surgical procedures to be performed without the use of antimicrobial prophylaxis. Thus, from 2019 to 2021, we eliminated the use of unnecessary prophylactic antibiotics for common research surgeries performed by our department. In this study, we compared complication rates of animals receiving PLN and laparoscopic biopsies (MLN, liver, and/or spleen), with and without the use of prophylactic antibiotics. A majority of animals was SIV or SHIV infected at the time of their procedure, so we also compared post-operative complication rates of infected and uninfected animals.

Similar to other investigators' findings, there were no significant differences in PLN biopsy complication rates between animals that received prophylactic antibiotics and those that did not [15, 21]. Several factors regarding surgical techniques may explain this result. Surgeons performing biopsies in this study maintained strict surgical asepsis, so all procedures were categorized as clean [1]. Additionally, surgeons received extensive training to perform the procedures, which included emphasis on gentle tissue handling, hemostasis, and intradermal suture placement. Indeed, aseptic technique and proper tissue handling should eliminate the need for prophylactic antibiotics in most sterile procedures [1]. Finally, most surgeons performed hundreds of PLN biopsies over the course of this study and were therefore highly proficient. This likely led to decreased intraoperative time, reducing risk of post-operative infection [22]. Therefore, because of good surgical technique and efficiency, prophylactic antibiotics were unnecessary and eliminating their use made no difference in post-operative complication rates.

**Table 3. Surgical complication rates based on infection status and treatment group.**

| Surgery Type | | Prophylaxis | No Prophylaxis | Total |
|---|---|---|---|---|
| Peripheral lymph node biopsies | | | | |
| | SIV/SHIV + | 2.64% (n = 455) | 2.15% (n = 885) | 2.31% (n = 1,340) |
| | SIV/SHIV − | 2.16% (n = 556) | 1.8% (n = 334) | 2.02% (n = 890) |
| Laparoscopic biopsies | | | | |
| | SIV/SHIV + | 5.13%* (n = 39) | 0.24%* (n = 822) | 0.46% (n = 861) |
| | SIV/SHIV − | 2.27% (n = 44) | 1.62% (n = 494) | 1.67% (n = 538) |

Complication rates of peripheral lymph node (PLN) and laparoscopic biopsy procedures in SIV/SHIV infected and uninfected animals, with and without use of prophylactic antibiotics. Asterisk (*) indicates statistically significant results ($p$ value <0.05).

Interestingly, for laparoscopic procedures, there were significantly less complications in animals that did not receive prophylactic antibiotics. It is unclear why this is the case. From 2018 to early 2019, prophylactic antibiotics were utilized for all laparoscopic biopsies. In early 2019, our department discontinued use of antimicrobial prophylaxis for these procedures altogether. It is possible that as surgical staffs' experience and efficiency increased over the course of this study, complication rates decreased, independent of antimicrobial prophylaxis use. However, some studies do show that antibiotic prophylaxis increases the risk of SSIs in humans [3, 16], so it is also possible that antibiotic use actually increased complication rates in the animals in this study.

SIV/SHIV infection did not appear to increase risk of surgical complications in this study. In fact, SIV/SHIV infected animals appeared to have a lower risk of complications following laparoscopic biopsies compared to uninfected animals, though this result was not significant. There are several explanations as to why SIV/SHIV infection did not increase post-operative complications. Some animals in this study were receiving combination anti-retroviral therapy (cART). Thus, they had low to undetectable viral loads and normal CD4+ cell counts at the time of surgery. Additionally, all study animals who progress to AIDS criteria reach study endpoint and are euthanized. Therefore, it is highly likely that the majority of animals who received biopsies were not immunosuppressed to a degree that would predispose them to SSIs at the time of the biopsy. Other studies have shown that the decline in CD4+ counts directly correlates with severity of immunocompromised state and susceptibility to SSI [5]. In a study examining post-operative complication rates in women after gynecologic surgery, CD4 + counts < 200/uL and advanced HIV infection were risk factors for complications [15]. Another study reviewing post-operative infection rates in HIV positive patients showed that patients with lower preoperative CD4+ counts were more likely to develop SSIs following abdominal surgery [16]. Finally, wound healing is not impaired in HIV positive patients [23], and is also likely unimpaired in SIV/SHIV infected macaques.

Antimicrobial prophylaxis had no significant effect on PLN biopsy post-operative complication rates in SIV/SHIV infected animals, and SIV/SHIV infected animals that received prophylactic antibiotics had higher complication rates post-laparoscopy. As previously discussed, good surgical technique, strict asepsis, and efficiency likely helped prevent SSIs in both SIV/SHIV infected and uninfected animals. Other factors may also have contributed, such as invasiveness of the procedure and surgical wound classification. HIV positive humans have a higher risk of SSI following extensive abdominal surgery than uninfected individuals, but have a similar risk when more minor procedures are performed, such as episiotomy or suturing of vaginal tears post-delivery [15]. In another study, HIV patients with dirty wounds had a 100% chance of SSI compared to 2.6% in HIV patients with clean wounds [16]. As animals in this study received clean, minimally invasive surgical biopsies rather than contaminated or major abdominal surgeries, their initial risk of SSI was relatively low and thus antimicrobial prophylaxis had no positive effect on post-operative complication rates.

Our department focused on eliminating inappropriate use of antimicrobial prophylaxis for a number of reasons, including improvement of animal health and welfare. Excessive use of antibiotics has selected for antimicrobial resistance in many bacteria, rendering many antibiotic treatments ineffective. Multi-drug resistant bacteria have emerged due to antibiotic misuse, with a consequent increase in mortality due to infectious diseases such as MRSA, *Clostridium difficile*, and VRE [24]. MRSA colonization is also quite common in research macaques, with colonization rates as high as 17.6% [25]. In a recent study of rhesus and cynomolgus macaques, previous antimicrobial usage was significantly linked to MRSA carriage [26].

Antibiotics may also negatively affect humans and animals in other ways, due to side effects and microbiome changes [8]. They have been shown to impact key target cell populations for SIV acquisition and disease progression in the colon, such as Th17+ CD4+ T cells [20], as well as inducing colitis, which has been shown to have a significant impact on acquisition and disease progression in SIV/SHIV models [23, 27, 28]. Even short courses of antibiotics can cause microbiome perturbations that can persist for weeks to months [29]. Alterations of the microbiome may increase risk of certain viral infections, including Human Immunodeficiency Virus (HIV), and modify the immune response to various vaccines [29]. Additionally, there are many reports that indicate a positive correlation between antibiotic treatment and weight gain in both human and animal studies [24].

In conclusion, we recommend eliminating unnecessary antibiotic use in study animals due to their potential impact on research models and their potential to promote antimicrobial resistance. Antimicrobials should not be used based on perceived risk, as is the case when prophylactic antibiotics are used in SIV infected macaques undergoing clean, minimally invasive surgeries. It is essential to follow evidence-based practices, as were performed in this study, to ensure antimicrobial use has clear benefits, such as those demonstrated with elimination of endemic gastrointestinal pathogens in macaques [27]. Finally, given the similarities in outcomes between SIV infected macaques and HIV infected patients, it is likely that where similar conditions exist (highly trained surgeons, clean surgeries, minimally invasive procedures, and the ability to achieve good aseptic technique), prophylactic antibiotics are also likely unnecessary in HIV patients.

## Author Contributions

**Conceptualization:** Cassandra Moats, Jeremy V. Smedley.

**Data curation:** Cassandra Moats, Kimberly Cook, George Lawrence.

**Formal analysis:** Kimberly Cook, George Lawrence.

**Investigation:** Cassandra Moats, Rachele M. Bochart, George Lawrence, Jeremy V. Smedley.

**Methodology:** Cassandra Moats, Kimberly Armantrout, Hugh Crank, Rachele M. Bochart, George Lawrence, Jeremy V. Smedley.

**Resources:** Michael K. Axthelm, Jeremy V. Smedley.

**Software:** George Lawrence.

**Supervision:** Cassandra Moats, Rachele M. Bochart, Michael K. Axthelm, Jeremy V. Smedley.

**Validation:** Cassandra Moats, Kimberly Cook, Kimberly Armantrout, Hugh Crank, Samantha Uttke, Kelly Maher, Jeremy V. Smedley.

**Writing – original draft:** Cassandra Moats, Kimberly Cook, George Lawrence.

**Writing – review & editing:** Michael K. Axthelm, Jeremy V. Smedley.

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
