## [Decision Letter · Decision Letter 0]

14 Dec 2021

PONE-D-21-26281

Antimicrobial prophylaxis does not improve post-surgical outcomes in SIV/SHIV-uninfected or SIV/SHIV-infected macaques (Macaca mulatta and Macaca fascicularis)

Dear Dr. Rae Moats, 

Thank you for submitting your manuscript to PLOS ONE. After careful consideration, we feel that it has merit but does not fully meet PLOS ONE’s publication criteria as it currently stands. Therefore, we invite you to submit a revised version of the manuscript that addresses the points raised during the review process.

Both reviewers feels there is a merit but raised several questions and suggestions. Therefore, please carefully review their comments and incorporate during revision. 

We look forward to receiving your revised manuscript.

Kind regards,

Siddappa N. Byrareddy, PhD

Academic Editor

PLOS ONE

2. Please upload a copy of Supporting Information Figure/Table/etc. s1 and s2 which you refer to in your text on page 26.

Reviewers' comments:

Reviewer's Responses to Questions

**Comments to the Author**

1. Is the manuscript technically sound, and do the data support the conclusions?

Reviewer #1: Yes

Reviewer #2: Yes

2. Has the statistical analysis been performed appropriately and rigorously? 

Reviewer #1: I Don't Know

Reviewer #2: Yes

3. Have the authors made all data underlying the findings in their manuscript fully available?

Reviewer #1: No

Reviewer #2: Yes

4. Is the manuscript presented in an intelligible fashion and written in standard English?

Reviewer #1: Yes

Reviewer #2: Yes

5. Review Comments to the Author

Reviewer #1: This manuscript discusses the use and appropriateness of prophylactic antimicrobials perioperatively in nonhuman primates and more specifically SIV/SHIV infected NHPs. This is a retrospective analysis of the institutions post-operative complication rate with and without the use of prophylactic antibiotics perioperatively. Their analysis of their records indicates that there is not a statistically significant difference between the use of antibiotics perioperatively and that there may be evidence that antibiotic use adversely impacts post-operative complication rates. Overall this information is useful especially in an animal model that is sometimes thought of as immunocompromised. This study is in support of the current practices and policies addressing responsible antibiotic stewardship that has become popular in human and veterinary medicine over the last 10 years. Overall I believe this publication is useful to the veterinary and laboratory animal community. My concerns are with the somewhat inflated relevance of the data and the inaccurate statements in the introduction and discussion indicating that this is somehow a novel concept and that antibiotics are commonly and indiscriminately used perioperatively in the veterinary community and in the laboratory animal community. Lines 63-65: “ There is a common perception among researchers and veterinary staff that NHPs require prophylactic antibiotics for surgical procedures, regardless of wound classification, surgery length, host status, or implant placement.” The authors provide little support for these statements in the literature within the last 10 years. The references provided indicating that antibiotics are commonly used are 10-20yrs old. The AVMA has put out guidelines on responsible antibiotic use (https://www.avma.org/antimicrobials-guidelines-judicious-therapeutic-use), the APV (https://www.primatevets.org/guidance-documents), the AAHA (https://www.aaha.org/aaha-guidelines/use-of-antimicrobials-configuration/judicious-therapeutic-use-of-antimicrobials-in-cats-and-dogs/), among many others. A quick internet search revealed the book "Nonhuman Primates in Biomedical Research: Biology and Management", Chapter 14, pg 351 states: "...Major emphasis must be put on using sterile or sterilized materials and aseptic surgical techniques....The routine use of prophylatic antibiotics, however, is a practice that should be considered based upon the likelihood of intraoperative or postoperative contamination and the complexity and duration of the surgery." Indiscriminate and standard use of antibiotics in nonhuman primates, whether SIV infected or not, is not common practice or novel.

Lines 86-87: "Therefore, it is likely that during many of these studies, prophylactic antibiotics were administered in the peri-operative period but were not described in the materials and methods sections.placement.” This statement is not an accurate assumption and should be reframed as the authors' anectdotal opinion.

Lines 219-231: Were CBCs or cultures performed or evaluated? NHPs are stoic and subclinical infections can go undetected. Evaluation of CBCs along with WBC and neutrophil distribution would have been useful information to determine the presence or absence of infection.

Lines 430-432: There is a previous publication by some of the authors included here advocating for multimodal antimicrobial use in the NHP SIV model ("Mitigation of endemic GI-tract pathogen-mediated inflammation through development of multimodal treatment regimen and its impact on SIV acquisition in rhesus macaques"). Given how strongly this publication advocates for not using antimicrobials in an SIV NHP model the contradictions of these publications should be addressed.

Lines 432-437: These sentences are either contradictory or not clear. One sentence indicates that due to the similarities between humans and NHPs that prophylactic antibiotics are not necessary while the following sentence seems to indicate that NHPs are the worse case scenario for evaluating the need for antibiotics. Which would seemingly justify clinicians concerns and decision to use antibiotics. This needs clarification or rewording.

Reviewer #2: Here authors want to examine whether antimicrobial prophylaxis influence the post surgical outcomes in SIV/SHIV-uninfected or SIV/SHIV-infected macaques. The study is important for surgery of patients or for research purpose. The findings will add some information in modern literature that deals with different surgical techniques. However, I have some concerns on the manuscripts.

Title: the title is to be modified. The authors should mention about the type of study: prospective or retrospective?

Abstract: The abstract should be brief (objective- methods-findings- final conclusion). Addition of excessive result data is not required

Introduction: page no 101-106: This section should be converted into generalized statements. It should not be restricted to ONPRC infectious disease facility as such type of surgeries are also practiced in other research centers of various countries.

Materials and methods

L 208-210: The results of this section should reflect on result section.

L237-241: authors have mentioned a list of complications. On what basis, authors identified the true complication? It is not clear.

Table 1 & 2: it is not clear how the authors calculated %, what about the raw data status?

Statistical analysis: Authors mostly describe about the software. Which statistical methods were used to examine the significant difference between the percentages?

A consort statement (study flow diagram) is required.

Result

L 257: Need a concise information about demographics of cases

The digit of P value should be uniform.

Throughout the result section, authors mentioned the no of observations repeatedly. It is not desirable. These should be depicted in table format. After that results should be presented as to- the- point.

Discussion:

L335-337: How antibiotics affect key immune parameters particularly cefazolin & cephalexin (L205-206) that were used in surgeries? Explain.

L 344-353: the repetitions of results in the discussion is not desirable.

L356-360/366-377: authors use some superfluous/redundant statements that does not required without suitable references.

Specific comments:

a) It may be true that antimicrobial prophylaxis had no significant effect on PLN biopsy post-operative complication rates in SIV/SHIV infected animals but suitable explanations are lacking here to support the statement “’ SIV/SHIV infected animals that received prophylactic antibiotics had higher complication rates post-laparoscopy”

b) In this study, true post-operative complication were identified as inappetence; moderate to severe inflammation, swelling or bruising; evidence of pain; incision discharge; surgical site abscessation; suture removal; incision ulceration; and/or incision dehiscence.

c) The Antimicrobial prophylaxis may influence the post operative inappetence by gut upset/ altering the gut microbiota as the authors mentioned in the manuscript. However, it not clear how antimicrobial prophylaxis influence the above mentioned other identified complications.

6. PLOS authors have the option to publish the peer review history of their article (what does this mean?). If published, this will include your full peer review and any attached files.

Reviewer #1: No

Reviewer #2: **Yes: **SRISHTI SONI

---

## [Author Response · Author response to Decision Letter 0]

8 Feb 2022

Further responses to the editor and reviewers can be found in our "Response to Reviewers" letter, which we have uploaded. However, I have included specific comments to each reviewer below.

Comments from Reviewer 1

• Comment 1: This manuscript discusses the use and appropriateness of prophylactic antimicrobials perioperatively in nonhuman primates and more specifically SIV/SHIV infected NHPs. This is a retrospective analysis of the institutions post-operative complication rate with and without the use of prophylactic antibiotics perioperatively. Their analysis of their records indicates that there is not a statistically significant difference between the use of antibiotics perioperatively and that there may be evidence that antibiotic use adversely impacts post-operative complication rates. Overall this information is useful especially in an animal model that is sometimes thought of as immunocompromised. This study is in support of the current practices and policies addressing responsible antibiotic stewardship that has become popular in human and veterinary medicine over the last 10 years. Overall I believe this publication is useful to the veterinary and laboratory animal community. My concerns are with the somewhat inflated relevance of the data and the inaccurate statements in the introduction and discussion indicating that this is somehow a novel concept and that antibiotics are commonly and indiscriminately used perioperatively in the veterinary community and in the laboratory animal community. Lines 63-65: “ There is a common perception among some researchers and veterinary staff that NHPs require prophylactic antibiotics for surgical procedures, regardless of wound classification, surgery length, host status, or implant placement.” The authors provide little support for these statements in the literature within the last 10 years. The references provided indicating that antibiotics are commonly used are 10-20yrs old. The AVMA has put out guidelines on responsible antibiotic use (https://www.avma.org/antimicrobials-guidelines-judicious-therapeutic-use), the APV (https://www.primatevets.org/guidance-documents), the AAHA (https://www.aaha.org/aaha-guidelines/use-of-antimicrobials-configuration/judicious-therapeutic-use-of-antimicrobials-in-cats-and-dogs/), among many others. A quick internet search revealed the book "Nonhuman Primates in Biomedical Research: Biology and Management", Chapter 14, pg 351 states: "...Major emphasis must be put on using sterile or sterilized materials and aseptic surgical techniques....The routine use of prophylatic antibiotics, however, is a practice that should be considered based upon the likelihood of intraoperative or postoperative contamination and the complexity and duration of the surgery." Indiscriminate and standard use of antibiotics in nonhuman primates, whether SIV infected or not, is not common practice or novel. 

Response: We thank the reviewer for their supportive comments and agree that there is a lot of support for the approach we are advocating in this manuscript. We have undertaken a retrospective review that provides data to support changes at institutions that, as noted by the reviewer, still look at things like the immunocompromised status of SIV/SHIV infected macaques as a rational to administer prophylactic antibiotics even for clean procedures. 

It is worth noting that in the past 10 years, it was standard practice for all SIV infected macaques at the Washington National Primate Research Center, the Oregon National Primate Research Center, and some NIH institutes to receive prophylactic antibiotics prior to all procedures described in this manuscript based on the authors’ experience at these institutions. There was significant resistance to change and the data presented in this manuscript should help others to overcome hurdles that we faced in eliminating unnecessary antibiotic use, which was still ongoing in 2019 at the ONPRC. All of these institutions constitute major primate research programs. Therefore, we agree with the reviewer that sharing this information will be helpful to the field. We have updated our wording as above in the revised manuscript.

• Comment 2: Lines 86-87: "Therefore, based on the author’s experience, it is likely possible that during many of these studies, prophylactic antibiotics were administered in the peri-operative period but were not described in the materials and methods sections.placement.” This statement is not an accurate assumption and should be reframed as the authors' anectdotal opinion.

Response: Thank you, we agree. We have updated this in the revised manuscript, as above.

• Comment 3: Lines 219-231: Were CBCs or cultures performed or evaluated? NHPs are stoic and subclinical infections can go undetected. Evaluation of CBCs along with WBC and neutrophil distribution would have been useful information to determine the presence or absence of infection. 

Response: CBCs and cultures were only performed if clinically indicated and are not part of the routine postoperative care. As this is a retrospective analysis we only have what was collected at the time of these procedures.

• Comment 4: Lines 430-432: There is a previous publication by some of the authors included here advocating for multimodal antimicrobial use in the NHP SIV model ("Mitigation of endemic GI-tract pathogen-mediated inflammation through development of multimodal treatment regimen and its impact on SIV acquisition in rhesus macaques"). Given how strongly this publication advocates for not using antimicrobials in an SIV NHP model the contradictions of these publications should be addressed.

Response: Thank you. We have updated the revised manuscript to address this potential contradiction.

• Comment 5: Lines 432-437: These sentences are either contradictory or not clear. One sentence indicates that due to the similarities between humans and NHPs that prophylactic antibiotics are not necessary while the following sentence seems to indicate that NHPs are the worse case scenario for evaluating the need for antibiotics. Which would seemingly justify clinicians concerns and decision to use antibiotics. This needs clarification or rewording. 

Response: We have omitted this last sentence and have reworded the entire last paragraph of the revised manuscript. 

Comments from Reviewer 2

• Comment 1: Here authors want to examine whether antimicrobial prophylaxis influence the post surgical outcomes in SIV/SHIV-uninfected or SIV/SHIV-infected macaques. The study is important for surgery of patients or for research purpose. The findings will add some information in modern literature that deals with different surgical techniques. However, I have some concerns on the manuscripts.

Title: the title is to be modified. The authors should mention about the type of study: prospective or retrospective?

Response: We have changed the title as suggested in the revised manuscript.

• Comment 2: Abstract: The abstract should be brief (objective- methods-findings- final conclusion). Addition of excessive result data is not required

Response: We have removed excessive results data from the abstract in the revised manuscript.

• Comment 3: Introduction: page no 101-106: This section should be converted into generalized statements. It should not be restricted to ONPRC infectious disease facility as such type of surgeries are also practiced in other research centers of various countries.

Response: We also agree and have re-worded into a more generalized statement in the manuscript.

• Comment 4: Materials and methods L 208-210: The results of this section should reflect on result section. 

Response: We thank the reviewer for pointing this out and have added a sentence clarifying how the SIV status was determined for the animals represented as infected or uninfected in the results section. 

• Comment 5: L237-241: authors have mentioned a list of complications. On what basis, authors identified the true complication? It is not clear. 

Response: A true complication was a post-operative surgical complication that either required medical or surgical treatment, and/or resulted in the need to increase the standard post-operative observation period to allow for further monitoring to ensure the animal did not require further treatment. This statement has been added to the manuscript, and this paragraph has been reworded to hopefully provide clarification.

• Comment 6: Table 1 & 2: it is not clear how the authors calculated %, what about the raw data status? 

Response: The raw data is the animal record, which we have pulled from our electronic record system into an excel format. We have provided raw data for your review. We also included a description of how the percentage was calculated in the manuscript.

• Comment 7: Statistical analysis: Authors mostly describe about the software. Which statistical methods were used to examine the significant difference between the percentages? 

Response: We used permutation tests (also known as randomization tests) to determine if our observed complication rates differed between treatment groups. This nonparametric method relies on an empirical null distribution built by randomly resampling data with the associations of interest broken by that random sampling. We have added some more language in the statistical analysis section which will hopefully clarify this method. 

• Comment 8: A consort statement (study flow diagram) is required. 

Response: We thank the reviewer for this excellent suggestion and have created a study flow chart to better illustrate study group allocation. This is now Figure 1.

• Comment 9: Result L 257: Need a concise information about demographics of cases

Response: Thank you, we agree as well. We have updated this section into a table format (Table 1) in the manuscript, which hopefully makes this more concise.

• Comment 10: The digit of P value should be uniform.

Response: The P value in the manuscript has been made uniform (to 3 digits after the decimal for all p values).

• Comment 11: Throughout the result section, authors mentioned the no of observations repeatedly. It is not desirable. These should be depicted in table format. After that results should be presented as to- the- point. 

Response: We agree, this was quite wordy (especially the table legends), so we have moved more of the details to the table itself to reduce the information in the legend.

• Comment 12: Discussion: L335-337: How antibiotics affect key immune parameters particularly cefazolin & cephalexin (L205-206) that were used in surgeries? Explain. https://www.ncbi.nlm.nih.gov/pmc/articles/PMC7183431/

Response: We have added further explanation to the revised manuscript discussion. Antibiotics can induce colitis and affect key immune parameters, such as increasing the numbers of significant target cells (Th17+ CD4 T cells) for SIV infection in the colon, which could negatively influence infectious disease models (Zevin 2017; Manuzak 2019). Based on this data, we convinced our institution to change its practice in 2019 and allow common surgical procedures to be performed without the use of antimicrobial prophylaxis. 

• Comment 13: L 344-353: the repetitions of results in the discussion is not desirable.

Response: We agree and have removed repetition of results in the revised manuscript.

• Comment 14: L356-360/366-377: authors use some superfluous/redundant statements that does not required without suitable references.

Response: We agree and have removed much of the wording in L366-377 in the revised manuscript. We have also added a reference to L356-360 to support a statement that required a reference.

• Comment 15: It may be true that antimicrobial prophylaxis had no significant effect on PLN biopsy post-operative complication rates in SIV/SHIV infected animals but suitable explanations are lacking here to support the statement “’ SIV/SHIV infected animals that received prophylactic antibiotics had higher complication rates post-laparoscopy”

Response: This statement reflects the result that the complication rate for SIV/SHIV infected animals was significantly higher (p=0.011) for those that received antibiotics (5.13%) vs those that did not (0.24%). Though, as this was a retrospective analysis, we do not have direct data to determine why this is the case, we have attempted to offer potential explanations (lines 366-379) in the discussion. 

• Comment 16: In this study, true post-operative complication were identified as inappetence; moderate to severe inflammation, swelling or bruising; evidence of pain; incision discharge; surgical site abscessation; suture removal; incision ulceration; and/or incision dehiscence 

Response: We have attempted to clarify this as noted above. A true complication was a post-operative surgical complication that either required medical or surgical treatment, and/or resulted in the need to increase the standard post-operative observation period to allow for further monitoring to ensure the animal did not require further treatment. This statement has been added to the manuscript, and the associated paragraph has been reworded to hopefully provide clarification. 

• Comment 17: The Antimicrobial prophylaxis may influence the post operative inappetence by gut upset/ altering the gut microbiota as the authors mentioned in the manuscript. However, it not clear how antimicrobial prophylaxis influence the above mentioned other identified complications. 

Response: In this study, we utilized a program to flag our electronic records system for scenarios that could indicate a post-operative surgical complication (using criteria described in the manuscript). Once the record system flagged these cases, we reviewed each case manually to determine if the complication was “true” or not – true complications either required medical or surgical treatment, and/or resulted in the need to increase the standard post-operative observation period beyond 8 days to allow for further monitoring to ensure the animal did not require further treatment. For all flagged cases, we reviewed all post-operative observation notes. For example, if a procedure was flagged because the post-operative case was open for 10 days (standard observation window is 7-8 days), and the post-operative observation notes included descriptions such as “moderate inflammation” or “discharge” of the surgical site, that case had a true post-operative complication. Other more non-specific observations such as post-operative inappetence, could be due to a variety of factors (ex. GI upset due to opiate or antibiotic administration, infection, pain, etc.) but were still considered a “true complication,” if they resulted in the need for NSAIDs, opiates, or antibiotics to be prescribed, or required veterinary staff to keep the case open for longer than the standard observation window to monitor the animal. In cases such as these, though the complication could be for the variety of reasons and not necessarily indicative of a surgical site infection, we did not omit such complications as we could not rule out that a subclinical SSI was not the cause of the complication. However, there were a number of cases flagged on initial review as post-operative complications due to other injuries (ex. social partner fight trauma) or common NHP illnesses (ex. Shigella enteritis) that happened to crop up within the post-operative observation period, as these necessitated NSAID/antimicrobial/opiate use during that window and/or required extending post-operative observation periods. Manual records review of these flagged cases revealed that these were not related to the surgery, and thus such cases were not considered “true complications.”

As a retrospective study, it is only possible to determine if the presence or absence of antimicrobial prophylaxis had a significant effect on post-operative complication rates, based on the data present in the animal’s records, not exactly how antimicrobials could impact each type of post-operative observation.

---

## [Decision Letter · Decision Letter 1]

24 Mar 2022

Antimicrobial prophylaxis does not improve post-surgical outcomes in SIV/SHIV-uninfected or SIV/SHIV-infected macaques (Macaca mulatta and Macaca fascicularis) based on a retrospective analysis

PONE-D-21-26281R1

Dear Dr. Rae Moats,

We’re pleased to inform you that your manuscript has been judged scientifically suitable for publication and will be formally accepted for publication once it meets all outstanding technical requirements.

Kind regards,

Siddappa N. Byrareddy, PhD

Academic Editor

PLOS ONE

Additional Editor Comments (optional):

Reviewers' comments:

Reviewer's Responses to Questions

**Comments to the Author**

1. If the authors have adequately addressed your comments raised in a previous round of review and you feel that this manuscript is now acceptable for publication, you may indicate that here to bypass the “Comments to the Author” section, enter your conflict of interest statement in the “Confidential to Editor” section, and submit your "Accept" recommendation.

Reviewer #1: All comments have been addressed

Reviewer #2: All comments have been addressed

2. Is the manuscript technically sound, and do the data support the conclusions?

Reviewer #1: Partly

Reviewer #2: Yes

3. Has the statistical analysis been performed appropriately and rigorously? 

Reviewer #1: I Don't Know

Reviewer #2: Yes

4. Have the authors made all data underlying the findings in their manuscript fully available?

Reviewer #1: Yes

Reviewer #2: Yes

5. Is the manuscript presented in an intelligible fashion and written in standard English?

Reviewer #1: Yes

Reviewer #2: Yes

6. Review Comments to the Author

Reviewer #1: None

Reviewer #2: All the comments have been thoroughly addressed.The author should be commended for carrying out such a significant research project. It is thorough and fully supports the researchers' recommendation. I strongly propose that it be published in PLOS ONE.

7. PLOS authors have the option to publish the peer review history of their article (what does this mean?). If published, this will include your full peer review and any attached files.

Reviewer #1: No

Reviewer #2: **Yes: **SRISHTI SONI

---

## [Editor Report · Acceptance letter]

28 Mar 2022

PONE-D-21-26281R1 

Antimicrobial prophylaxis does not improve post-surgical outcomes in SIV/SHIV-uninfected or SIV/SHIV-infected macaques (*Macaca mulatta* and *Macaca fascicularis*) based on a retrospective analysis 

Dear Dr. Moats:

I'm pleased to inform you that your manuscript has been deemed suitable for publication in PLOS ONE. Congratulations! Your manuscript is now with our production department. 

Kind regards, 

on behalf of

Dr. Siddappa N. Byrareddy 

Academic Editor

PLOS ONE